# A Natural Technology for Vacuum-Packaged Cooked Sausage Preservation with Potentially Postbiotic-Containing Preservative

**Aloizio Lemos de Lima** [1,2,*], **Carlos Alberto Guerra** [3], **Lucas Marques Costa** [3], **Vanessa Sales de Oliveira** [1], **Wilson José Fernandes Lemos Junior** [4], **Rosa Helena Luchese** [1] and **André Fioravante Guerra** [5,*]

1   Department of Food Technology, Federal Rural University of Rio de Janeiro, Seropédica 23897-970, Brazil; vanessasdo@bol.com.br (V.S.d.O.); rhluche@gmail.com (R.H.L.)
2   Instituto Federal do Rio de Janeiro (IFRJ), Pinheiral 27197-000, Brazil
3   BRC Ingredientes Ltda., Rio Claro 13505-600, Brazil; carlos@brcingredientes.com.br (C.A.G.); lucas@brcingredientes.com.br (L.M.C.)
4   Faculty of Science and Technology, Free University of Bolzano-Bozen, 39100 Bolzano, Italy; juniorjflemos@gmail.com
5   Departamento de Engenharia de Alimentos, Centro Federal de Educação Tecnológica Celso Suckow da Fonseca (CEFET/RJ), Valença 27600-000, Brazil
*   Correspondence: aloizio.lima@ifrrj.edu.br (A.L.d.L.); andre.guerra@cefet-rj.br (A.F.G.); Tel.: +55-21-98174-7619 (A.L.d.L.); +55-21-99142-3932 (A.F.G.)

**Abstract:** In this study, a potentially postbiotic-containing preservative (PPCP) was produced in an axenic fermentation system with *Lacticaseibacillus paracasei* DTA 83 as a natural technology alternative for vacuum-packaged cooked sausage preservation. Cooked sausage-related microorganisms were obtained during the induced spoiling process in packages by pair incubation of sausages at different temperatures. The turbidity method was used to determine the microbiota susceptibility to PPCP. A controlled in situ design was performed by adding PPCP on the surface or to the mass of the sausages. Sodium lactate FCC85, which was used according to the manufacturer's recommendation, was included in the design for comparison. The results revealed that PPCP was as efficient as FCC85, which indicates PPCP as a promising alternative to the use of natural technologies to preserve and develop functional cooked sausages. Moreover, a strategy to use preservatives in vacuum-packaged cooked sausages was presented: the concentration needed to achieve the total inhibition of the microbiota determined by an in vitro trial should be respected when adding PPCP on the sausages' surface. When adding PPCP to the mass of the sausages, the concentration that showed a partial inhibition in vitro can also be applied in situ.

**Keywords:** biocontrol; biocin; heat-inactivated microorganism; food safety; sustainability

## 1. Introduction

Lactic acid bacteria (LAB) constitute a heterogeneous group that has extensively reported on in the literature because of its potential benefits for consumer health [1,2]. *Lacticaseibacillus paracasei* DTA 83 has been described as a candidate strain to deliver probiotics in food matrices [3–5]. In contrast, since microorganisms may present invasive potential, studies have shown the administration of viable cells by healthy people as a subject of great concern. Thus, the use of postbiotics may be highlighted as a suitable alternative.

The presence of spoilage microorganisms in food represents a critical issue with repercussions on massive food waste and food loss worldwide [6]. The safety and stability of food may be affected by numerous factors, such as microbial presence and/or activity; biochemical, physical–chemical, and sensory alterations; nutritional losses; and others. When intrinsic and extrinsic aspects of food allow microbial growth, the microbial spoilage pathway becomes dominant [7,8].

Biopreservation is an alternative food preservation technology applied to replace artificial preservatives. The US Food and Drug Administration (FDA) recognizes both probiotics and their metabolites as Generally Recognized as Safe (G.R.A.S) (Section 2.1 CFR184). Thus, they are useful for controlling the development of pathogens and spoilage microorganisms in food and foodstuff. Moreover, FDA has determined that conditions for their use are prescribed in the referent regulations and are predicated on the use of nonpathogenic and nontoxicogenic strains of the respective organisms and on the use of current good manufacturing practice (184.1(b)). Despite all the advantages, the use of bacteriocins is still limited because of the high cost of their isolation and purification, mainly when considering their application in products of low cost. In this context, the use of precultured broth mediums by LAB, without bacteriocin isolation and purification, may be a promising strategy to prevent spoilage in meat products [9].

Heat-treated meat products, such as vacuum-packaged cooked sausages, are traditionally marketed at room temperature in Brazil, leading to food waste due to spoilage processes that may occur before the shelf life determined by the manufacturer [10]. The vacuum atmosphere selectively suppresses the growth of specific microbial groups, attributing the initial microbiota to anaerobic and facultative groups [11]. These microorganisms overgrow and produce metabolites that cause the rejection of the products by consumers [12]. As a solution, the food industry often increases the concentration of preservatives in meat products, which may result in abusive use.

Sodium lactate is a widespread commercial preservative commonly used in sausages to control microbial growth and increase shelf life [13]. However, the higher the concentration of sodium lactate added to a food product, the higher the content of sodium. Therefore, although sodium lactate is a safe preservative for food and foodstuff, its excessive intake may result in increased blood pressure for consumers [14]. Indeed, natural technologies to preserve food are of growing interest to food industries and consumers.

Metabolites produced by LAB have been extensively tested to biocontrol the growth of pathogenic and spoilage microorganisms in meat ecosystems [15–17]. *L. paracasei* DTA 83 is a strain of human origin of great functional and technological interest. It is a ɤ-hemolytic and nonantibiotic resistant strain. Previous studies have demonstrated its potential to control the growth of *Escherichia coli*, *Salmonella Typhimurium*, *Listeria innocua*, and *Candida albicans* even after partial reduction in cell viability due to stress in the gastrointestinal transit. Technological features associated with the ability of *L. paracasei* DTA 83 to assimilate sugar in hardship conditions, such as brewer wort and plant extract solutions, were presented by Silva et al. and Oliveira et al. [18,19]. These aspects were decisive for selecting the strain for bioproduct processing. Moreover, maternal supplementation with *L. paracasei* DTA 83 reduced the expression of GAD 65, GAD 67, and GABAA receptor $\alpha3$ subunit in the hippocampus, modulating Swiss mice offspring [4].

Thus, this study aimed to compare the efficacy of potentially postbiotic-containing preservative (PPCP) produced by an axenic fermentation system with *L. paracasei* DTA 83 and sodium lactate in extending the use-by date of vacuum-packaged cooked sausages. Moreover, a strategy based on the co-use of preservative and cold chain management was presented to retain the original properties of the sausages during the proposed shelf-life period of 90 days.

## 2. Materials and Methods

### 2.1. Microbial Collection and Inoculum Preparation

*L. paracasei* DTA 83 was isolated from newborns' stools at Rio de Janeiro (Brazil) in selective Lawvab agar medium as reported by Lemos Junior et al. [20]. The strain was genotypically identified by sequencing of the 16S rDNA region and clustered by genetic similarity with other *Lacticaseibacillus* strains of the collection (Figure S1a) [21]. Furthermore, the complete genome data was deposited in GenBank under the accession number QRBH00000000 [22]. The strain has been classified as G.R.A.S. and characterized as a potential probiotic according to Tarrah et al. and Laureano-Melo et al. [3,4]. The technological

features of the strain were assessed in food matrices by Silva et al. (Figure S1b) [5,18]. Additionally, it was described as a potential strain for delivering postbiotic compounds by Oliveira et al. [18].

*L. paracasei* DTA 83 cultures were thawed at 7 °C for approximately 4 h and centrifuged at 6000× *g* for 5 min (2K15, Sigma Laborzentrifugen, Osterode am Harz, Germany) for pellet separation. The liquid fraction was discarded. Then, the remaining cell pellet was reconstituted with MRS broth and then incubated overnight at 36 °C for the microbial growth. To obtain sufficient biomass to produce PPCP on a pilot-industrial scale, the cultures were scaled up 1/10 (*vol/vol*) at 36 °C in an axenic cultivation in a sterile MRS broth medium prepared with food-grade ingredients to obtain 30 L of inoculum.

### 2.2. PPCP Production

A stirred tank bioreactor of 300 L, with automatic control of temperature and pH, was used to produce PPCP in an axenic fermentation system with *L. paracasei* DTA 83. This part of the experiment was carried out at BRC Ingredientes Ltda., located in the city of Rio Claro, state of São Paulo, Brazil. Modified MRS broth was prepared with food-grade ingredients without the addition of polysorbate 80 (Tween 80). The heat treatment was performed in a tank (heating up of 1 °C per minute) by the electrical activation of three resistors (3 kW). During heating, the medium was axially agitated at 84 rpm. The binomial 75 °C per 2 h was used to reduce the contaminants to an acceptable level (ca. 3 log cfu/g) and provide a competitive advantage to *L. paracasei* DTA 83 during the fermentation. After the heat treatment, the temperature of the medium was reduced to 36 °C (heating down of 0.5 °C per minute). *L. paracasei* DTA 83 biomass was produced in laboratory, scaling up 1/10 (*vol/vol*) of the culture into sterile modified MRS broth. A biological oxygen demand was used for incubation at 36 °C to obtain 30 L of inoculum. A culture with 18 h of growth, comprehended into the growth (log) phase, was added (1/10 of inoculum) into the bioreactor containing 270 L of modified MRS medium to obtain a final inoculum concentration of ca. 7 log cfu/mL. After 72 h of fermentation coupled with a pH decay to around 3.5, the medium was heat treated at 95 °C for 5 min (heating up of 1 °C per minute). PPCP was hot bottled in polypropylene containers of 10 L. The presence of remaining cells of *L. paracasei* DTA 83 or contaminants was assessed by plate counting on MRS and plate count agar and potato dextrose agar acidified to pH 3.5 with tartaric acid (all media from HiMedia, Mumbai, India).

### 2.3. In Vitro Efficacy of PPCP

Cooked sausage-related microorganisms were obtained from five packages of sausages, with collection at zero time (*n* = 1) and after pair incubation of samples at 7 °C (collection on days 3 and 6) and 36 °C (collection on days 2 and 4). A decimal suspension was prepared by weighing the sausages and adding 0.1% of peptone sterile water to the package. This step was conducted to count the microorganisms in the sausages, as well as those accumulated in the liquid inside the package after syneresis. After the samples were homogenized in a stomacher (SP-190, SPLabor, Brazil) for 90 s at 230 revolutions per minute (rpm), aliquots (100 μL) was transferred to tubes with 5 mL of brain–heart infusion, Casoy, deMan, Rogosa, and Sharp, and yeast–peptone–dextrose extract. The tubes were incubated at 36 °C for 24–48 h. The inoculum was obtained separately from each culture medium by transferring 1 mL of the tube content, with expressive growth (turbidity above 0.5 MacFarland standard), to an empty sterile screw-cap tube. Cells free of toxic compounds were obtained by twice washing the biomass cell pellets with a routine of centrifugation at 6000× *g* for 6 min for pellet sedimentation at the bottom of the tube, discarding the liquid fraction, adding 2 mL of phosphate buffer pH 7.2, and homogenizing in vortex. The turbidity of the microbial suspension was adjusted to 0.5 McFarland standard and 2-fold diluted. PPCP was randomly outlined to final concentrations of 0.0, 0.1, 0.3, 0.5, 1.0, 1.5, 2.0, 2.5, 3.0, and 3.5% (*vol/vol*) in the brain–heart infusion broth. The dilutions were prepared in the same media used in the test to avoid a shortage of nutrients for microbial growth. Finally,

100 µL of the microbial suspension was added into the tubes to achieve a final microbial concentration of ca. 5 log cfu/mL. A digital stirred water bath (SP-156/22, SPLabor, Brazil), with automatic temperature control, was used to incubate the tubes at 36 °C for 72 h. The absorbance was read in a photometry device at 600 nm (Spectrum SP-2000UV/2000UVPC, Shanghai, China) for a regular 6 h period. Before reading, the tubes were vortexed, and the absorbance was directly measured in the tubes. A tube without inoculum was used as blank and for equipment calibration at each reading.

### 2.4. In Situ Efficacy of PPCP

PPCP was tested in situ at concentrations of 1.0, 2.0, and 3.0% by adding the preservatives on the surface or to the mass of the sausages. The sausages were manufactured on an industrial-pilot scale for the meat industry located in Rio de Janeiro, Brazil. The production was performed according to the meat products' standard procedures, as follows: input of feed of raw materials, defrosting or breaking in frozen block crusher, grinding through industrial grinder knife (8–12 mm hole diameter (Ø) plate) (PC 106, Canoas, Brazil), mixing and addition of food ingredients (lean meat, pork fat, spices, and food additives) (250L, Cataguases, Brazil), stuffing in 15 and 250 mm inner (diameter × length) natural pork casing (NDX 22 Viscofan, Spain), cooking to achieve 72 °C (approximately 2 h) (MECA2G, Pará de Minas, Brazil) at the coldest point of the sausage, cooling by immersion in a cold water bath, and packing using a vacuum-package system with 5 to 7 pieces of sausage per package. PPCP was added to the mass of the sausages with other ingredients during the sausage mass preparation or directly into the packages to hurdle microbial growth after syneresis. The net weight of the sausages in the packages was used to calculate the volume of PPCP added into the packages. Sodium lactate FCC85 (Corbion, Purak, Brazil), added to the mass or on the sausages' surface, was included in the design to compare the efficacy of the PPCP with that of a reference widespread commercial preservative. The addition of sodium lactate was performed following the manufacturer's recommendation. Sausages without preservatives or with sterile deionized water, added to the mass or on the sausages' surface, were included as blank and control, respectively (Table 1). After manufacturing, the packages were immediately addressed to the laboratory.

**Table 1.** Formulation of pork sausage samples.

| | | | | | | | | | | |
|---|---|---|---|---|---|---|---|---|---|---|
| | **Treatments** | | | | | | | | | |
| | | **Sausage Surface** | | | | | | **Sausage Mass** | | |
| **Ingredients (%)** | **Blank** | **Control (Water) 2.0%** | **Sodium Lactate 2.0%** | **PPCP [3] 1.0%** | **PPCP 2.0%** | **PPCP 3.0%** | **Control (Water) 2.0%** | **Sodium Lactate 2.0%** | **PPCP 1.0%** | **PPCP 2.0%** | **PPCP 3.0%** |
| Lean pork meat | 67.33 | 67.33 | 67.33 | 67.33 | 67.33 | 67.33 | 67.33 | 67.33 | 67.33 | 67.33 | 67.33 |
| Pork fat | 20.00 | 20.00 | 20.00 | 20.00 | 20.00 | 20.00 | 20.00 | 20.00 | 20.00 | 20.00 | 20.00 |
| Drinking water | 10.00 | 10.00 | 10.00 | 10.00 | 10.00 | 10.00 | 8.00 | 8.00 | 9.00 | 8.00 | 7.00 |
| Salt (sodium chloride) | 1.80 | 1.80 | 1.80 | 1.80 | 1.80 | 1.80 | 1.80 | 1.80 | 1.80 | 1.80 | 1.80 |
| Seasoning [1] | 0.30 | 0.30 | 0.30 | 0.30 | 0.30 | 0.30 | 0.30 | 0.30 | 0.30 | 0.30 | 0.30 |
| Sodium trypoliphosphate | 0.32 | 0.32 | 0.32 | 0.32 | 0.32 | 0.32 | 0.32 | 0.32 | 0.32 | 0.32 | 0.32 |
| Sodium erythorbate | 0.10 | 0.10 | 0.10 | 0.10 | 0.10 | 0.10 | 0.10 | 0.10 | 0.10 | 0.10 | 0.10 |
| Curing salt [2] | 0.15 | 0.15 | 0.15 | 0.15 | 0.15 | 0.15 | 0.15 | 0.15 | 0.15 | 0.15 | 0.15 |
| Sterile deionized water | | 2.00 | | | | | 2.00 | | | | |
| Sodium lactate FCC85 | | | 2.00 | | | | | 2.00 | | | |
| PPCP | | | | 1.00 | 2.00 | 3.00 | | | 1.00 | 2.00 | 3.00 |

[1] Garlic powder, onion powder, black pepper, nutmeg, laurel powder, and celery powder; [2] sodium chloride (90%), sodium nitrite (6%), and sodium nitrate (4%); [3] potentially postbiotic-containing preservative.

### 2.5. Sample Characterization

2.5.1. Physicochemical Analyses

The analyses were carried out following the AOAC procedures [23]. Moisture content (%*w/v*) was determined by oven drying at 105 °C until constant weight. Ash content (%*w/v*) was determined by incinerating samples in a muffle furnace at 550 °C for 4 h.

Protein level (%$w/v$) was obtained by the Kjeldahl method. The Soxhlet extraction method with hexane was applied to determine the total fat content (%$w/v$). The total carbohydrate content was calculated as the difference between 100 and the sum of the percentages of moisture, ash, lipid, and protein. Total energy (kcal/100 g sample) was calculated according to the Atwater specific factor system (4.27 kcal/g for protein or carbohydrate and 9.02 kcal/g for fat).

### 2.5.2. Water Activity Measurement

Changes in the electrical conductivity of an electrolyte, in accordance with the method ISO 18787 (2017) [24], were used for water activity measurement in a AcquaLab Lite device (Decagon, Washington, USA) provided with a dielectric humidity sensor and infrared sample surface temperature. Before measuring, the equipment was calibrated with two standard solutions (K$_2$SO$_4$, *aw* 0.973 (CAS 7778–80–5) and KCl, *aw* 0,843 (CAS 7447–40–7)) provided by the manufacturer. A maximum error of $\pm0.005$ was considered as accuracy. To obtain a uniform sample, a piece of sausage was ground in an electric meat grinder (Centrífuga 1000, Britânia, Brazil). Excessive milling, which could lead to heating of samples and affect measurements, was avoided. Immediately after grinding, the sample portion was taken as quickly as possible to minimize exposure to humidity in the laboratory. A sample dish with a capacity of 7 mL was $^1/_3$ filled with sample so that there was no empty space at the bottom. During the analytical series, the measurement stability was verified using standard solutions. A waiting time of approximately 15 min was established between each measurement after opening the equipment lid.

### 2.5.3. pH Values

Nondestructive measurement of pH was performed according to the method ISO 2917:1999(E) [25]. A portable meat pHmeter device (pH Classic, Akso, Brazil), equipped with a knife probe electrode (IP65, Akso, Brazil) and automatic compensation of temperature, was used. Sausages were randomly withdrawn from the packages, and the pH value was determined by direct sticking the electrode in 3 different positions of the sausage: the two ends and the central section of the pieces. Before measuring, the equipment was calibrated with buffer solutions, pH 4.00 and pH 6.88 at 20 °C. A maximum error of $\pm0.01$ was considered as accuracy.

### 2.6. Durability Study

A predictive microbial method, named *MicroLab_ShelfLife*, was used to estimate the use-by date of vacuum-packaged cooked sausages at a chosen dynamic temperature profile (Figures S4 and S5 in Supplementary Material). The use-by dates for vacuum-packaged cooked sausages were established when spoilage microbial load achieved the maximum limit of ca. 9.3 log cfu/g. This is the borderline to determine when changes in sensory attributes related to the appearance of vacuum-packaged cooked sausages occur (Figures S2 and S3, Table S1). The horizontal method for enumeration of microorganisms (ISO 4833-1:2013) [26] was performed to determine the total microbial load, using plate count agar medium (HiMedia, Mumbai, India), at the zero time and after stimulating the microbial growth in the packages by pair incubation of samples at 7 and 36 °C, with counts on days 3 and 6 (7 °C) and on days 2 and 4 (36 °C) of incubation (Figure S4). The number of colonies obtained at each dilution level was imputed in the *MicroLab_ShelfLife* computational package to determine the parameters of the microbial growth and to plot the predictive microbial growth curve (Figure S5).

A dynamic temperature profile was entered in the predictive model based on the measurements published by the AccuWeather forecast during 2021. Latitude and longitude coordinates (22°54′13″ South; 43°12′35″ West; Rio de Janeiro, Brazil) were considered as the climatic location, indicating the place where the sausages would be sold. According to the Köppen–Geiger classification, the climate of Rio de Janeiro is a tropical monsoon climate (*Am*) [27]. The temperature data were grouped by season. The daily temperature

profile, representing each climate season, was hourly grouped to fit in the *MicroLab_ShelfLife* platform (Figure 1). This profile was used to mimic the temperature during the product storage and disposal for sale in markets.

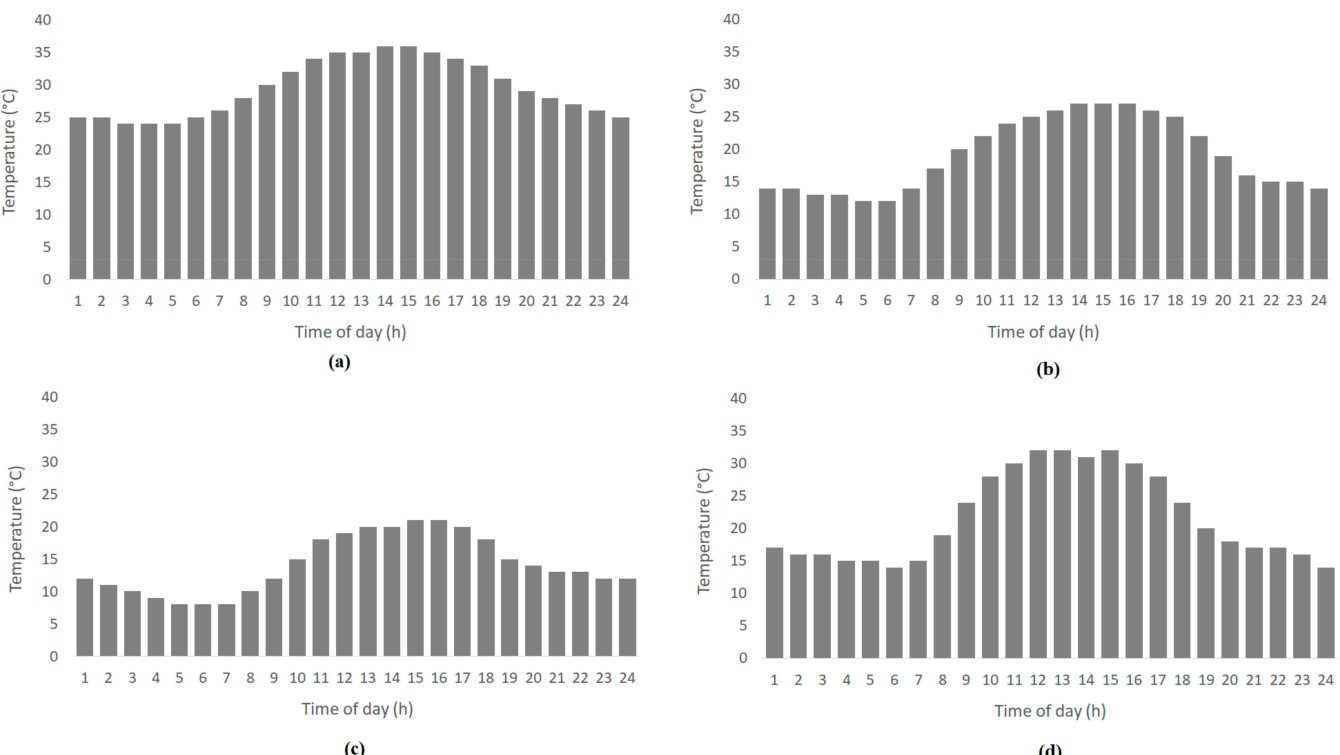

**Figure 1.** Temperature profile based on hourly variation during a one-day period to simulate the seasons: (**a**) summer, (**b**) autumn, (**c**) winter, (**d**) spring. They were determined based on the measurements published by AccuWeather (www.accuweather.com) for 2021. Latitude and longitude coordinates: 22°54′13″ South; 43°12′35″ West; Rio de Janeiro, Brazil, where a tropical monsoon climate (*Am*) has been reported (Köppen–Geiger climatic classification) [27].

### 2.7. Statistical Analyses

Results related to PPCP efficacy against the growth of natural microbiota and physico-chemical characterization of vacuum-packaged cooked sausages were obtained in triplicate. Linear regression was applied in the turbidity method regarding the incubation time with representative microbial growth, comprehended from 6 to 30 h of incubation at 36 °C, according to Equations (1)–(3). Angular coefficients from regressions (mean ± standard error) were evaluated by analysis of variance (ANOVA) followed by Fisher's (LSD) test ($p < 0.05$):

$$\dot{Y} \pm \frac{t\alpha}{2} \times SE \times \sqrt{hi} \tag{1}$$

$$\dot{Y} \pm t\alpha/2 \times SE \times \sqrt{1 + hi} \tag{2}$$

$$hi = 1/n + (xi - x)^2 / \sum (xi - x)^2 \tag{3}$$

$\dot{Y}$ is the estimated value, $t\alpha/2$ is the value of Student's t distribution, $n$ is the number of observations, $xi$ is the value of the sample, and $x$ is the mean.

A computational predictive modeling package, *MicroLab_ShelfLife*, was developed in the present study and used to predict the use-by date of vacuum-packaged cooked sausages from five packages of each sample group (Supplementary Material). The method was also used to evaluate the effect of the temperature associated with preservatives in the shelf life of the products and to estimate the initial microbial load of vacuum-packaged cooked sausages to achieve the proposed shelf-life period of 90 days.

### 3. Results

In vitro trials revealed that PPCP addition at concentrations up to 0.5% did not inhibit microbial growth. In samples containing 1.0–3.0% of PPCP, microbial inhibition was partially achieved. Although the efficacy was directly proportional to the added concentration of PPCP, similar results were obtained by adding 1.0 or 1.5% of PPCP ($p > 0.05$). Total inhibition was achieved at concentrations above 3.0% (Table 2, Figures 2 and 3).

**Table 2.** Linear regression parameters of microbial growth.

| | (%) of Potentially Postbiotic-Containing Preservative (PPCP) | | | | | | | | | |
|---|---|---|---|---|---|---|---|---|---|---|
| Coefficients | 0.0 | 0.1 | 0.3 | 0.5 | 1.0 | 1.5 | 2.0 | 2.5 | 3.0 | 3.5 |
| $xi$ | 0.044 [a] | 0.044 [a] | 0.044 [a] | 0.044 [a] | 0.035 [a] | 0.031 [a] | 0.025 [b] | 0.016 [c] | 0.004 [d] | 0.004 [d] |
| $yi$ | −0.299 | −0.294 | −0.298 | −0.305 | −0.309 | −0.281 | −0.242 | −0.137 | −0.018 | −0.034 |
| $R^2$ | 0.978 | 0.981 | 0.979 | 0.977 | 0.959 | 0.955 | 0.921 | 0.961 | 0.881 | 0.088 |
| SE | 0.111 | 0.109 | 0.110 | 0.111 | 0.088 | 0.079 | 0.065 | 0.040 | 0.012 | 0.011 |
| SQ | 1890 | 1890 | 1890 | 1890 | 1890 | 1890 | 1890 | 1890 | 1890 | 1890 |
| n | 18 | 18 | 18 | 18 | 18 | 18 | 18 | 18 | 18 | 18 |
| DF ($n - 2$) | 16 | 16 | 16 | 16 | 16 | 16 | 16 | 16 | 16 | 16 |
| $t\alpha/2$ | 2.4729 | 2.4729 | 2.4729 | 2.4729 | 2.4729 | 2.4729 | 2.4729 | 2.4729 | 2.4729 | 2.4729 |
| Confidence Interval | 0.95 | 0.95 | 0.95 | 0.95 | 0.95 | 0.95 | 0.95 | 0.95 | 0.95 | 0.95 |

$xi$—angular coefficient; $yi$—linear coefficient; $R^2$—coefficient of determination; SE—Standard error; SQ—sum of squares; n–number of observations; DF—degrees of freedom; $t\alpha/2$—value of Student's t distribution. Different capital letters indicate significant differences by Fisher's (LSD) test ($p < 0.05$).

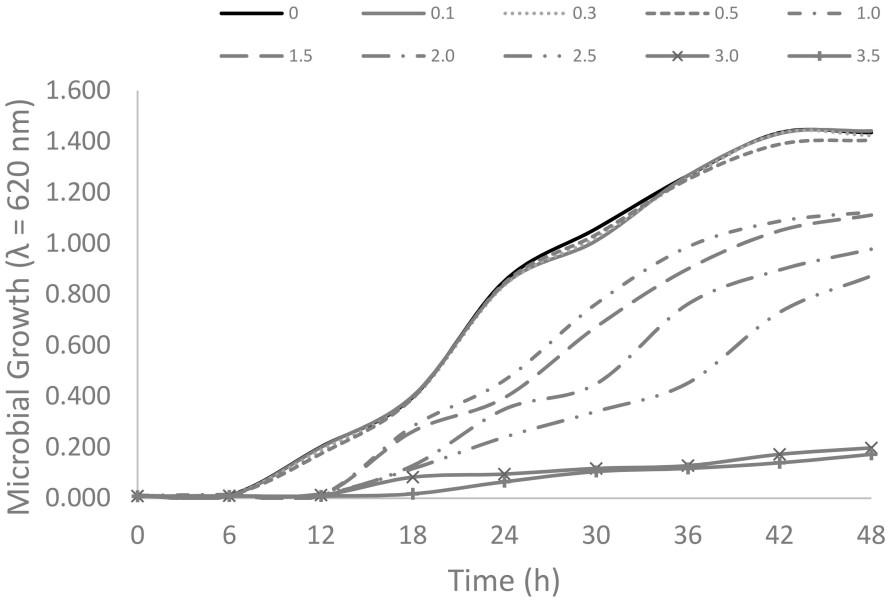

**Figure 2.** In vitro efficacy of PPCP against the growth of natural microbiota in vacuum-packaged cooked sausages. The inoculum was adjusted to ca. 5.5 log cfu/g before testing, and the turbidity method was used to evaluate the efficacy.

Table 3 shows the physicochemical characterization, water activity measurements, and pH values of sausages.

PPCP and FCC85 can reduce the growth of natural microbiota in vacuum-packaged cooked sausages and extend the shelf-life period. However, the strategy of addition must be carefully designed. The superficial treatments with 1.0% of PPCP and 2.0% of FCC85 should be discouraged, since these treatments did not present effective results compared with blank and control. In the sausages' mass, the addition of 1.0% PPCP was as effective as the addition of 2.0% of FCC85, indicating a potential natural alternative for product preservation (Table 4).

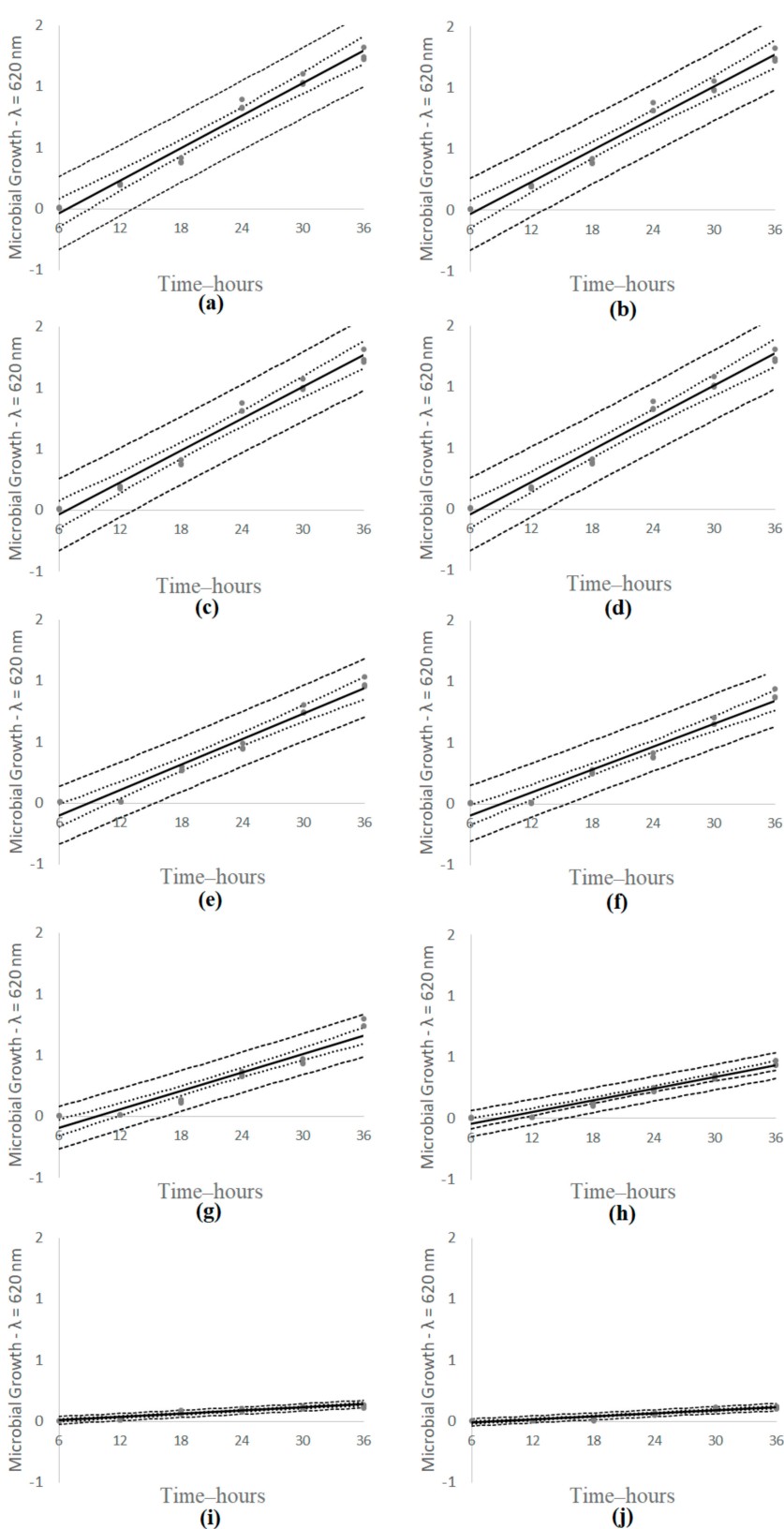

**Figure 3.** Linear regression (—-), confidence interval for the mean ( . . . .), and prediction interval for the sample (—) of the period with microbial growth (from 6 to 36 h of incubation at 36 °C) at different concentrations of potentially postbiotic-containing preservative (PPCP): (**a**) 0.0%; (**b**) 0.1%; (**c**) 0.3%; (**d**) 0.5%; (**e**) 1.0%; (**f**) 1.5%; (**g**) 2.0%; (**h**) 2.5%; (**i**) 3.0%; (**j**) 3.5%.

**Table 3.** Physicochemical characterization of vacuum-packaged cooked sausages (*n* = 3).

| Parameter | Mean ± Standard Error | |
|---|---|---|
| Moisture (%) | 56.663 | ± 0.160 |
| Protein (%) | 14.434 | ± 0.288 |
| Fat (%) | 23.550 | ± 0.122 |
| Ash (%) | 3.550 | ± 0.387 |
| Carbohydrates [1] (%) | 1.803 | ± 0.627 |
| Total energy (kcal/100 g sample) | 281.749 | ± 1.714 |
| Potential of hydrogen (pH) | 6.878 | ± 0.004 |
| Water activity (*Aw*) | 0.964 | ± 0.002 |

[1] Calculated according to the Atwater specific factor system (4.27 kcal/g for protein or carbohydrate and 9.02 kcal/g for fat).

**Table 4.** Durability study of vacuum-packaged cooked sausage samples.

| | | | | | | | | | | | | |
|---|---|---|---|---|---|---|---|---|---|---|---|---|
| | | | | | | | | **Treatments** | | | | |
| | **Sample Incubation** | | | **Sausage Surface** | | | | | **Sausage Mass** | | | |
| | Temperature (°C) | Time (days) | Blank | Control (water) 2.0% | Sodium lactate 2.0% | PPCP [4] 1.0% | PPCP 2.0% | PPCP 3.0% | Control (water) 2.0% | Sodium lactate 2.0% | PPCP 1.0% | PPCP 2.0% | PPCP 3.0% |
| Laboratorial data (log cfu/g) | | 0 | 5.77 | 5.80 | 5.71 | 5.88 | 5.79 | 5.80 | 5.66 | 5.76 | 5.81 | 5.83 | 5.87 |
| | 7 | 3 | 6.01 | 5.99 | 5.89 | 5.99 | 5.90 | 5.81 | 5.95 | 6.01 | 6.02 | 5.92 | 6.01 |
| | | 6 | 6.49 | 6.48 | 6.37 | 6.45 | 6.23 | 5.98 | 6.42 | 6.14 | 6.09 | 6.00 | 6.32 |
| | 36 | 2 | 6.69 | 6.72 | 6.70 | 6.80 | 6.64 | 6.14 | 6.59 | 6.66 | 6.61 | 6.33 | 6.26 |
| | | 4 | 8.50 | 8.48 | 8.49 | 8.61 | 8.45 | 6.87 | 8.42 | 7.12 | 7.10 | 6.97 | 6.94 |
| Specific maximum growth rate (log cfu/g/day) | 7 | L phase | 0.1000 | 0.0883 | 0.0850 | 0.0658 | 0.0550 | 0.0167 | 0.1117 | 0.0733 | 0.0583 | 0.0292 | 0.0608 |
| | | D phase | 0.0287 | 0.0253 | 0.0244 | 0.0189 | 0.0158 | 0.0048 | 0.0320 | 0.0210 | 0.0167 | 0.0084 | 0.0174 |
| | 36 | L phase | 0.5713 | 0.5650 | 0.5950 | 0.5713 | 0.5450 | 0.2188 | 0.5775 | 0.3950 | 0.3613 | 0.2675 | 0.2313 |
| | | D phase | 0.1637 | 0.1619 | 0.1705 | 0.1637 | 0.1562 | 0.0627 | 0.1655 | 0.1132 | 0.1035 | 0.0767 | 0.0663 |
| | | Season | | | | | | | | | | | |
| Ngrowth (log cfu/g/day) [1] | | Summer | 0.4649 | 0.4575 | 0.4800 | 0.4572 | 0.4345 | 0.1732 | 0.4724 | 0.3224 | 0.2929 | 0.2137 | 0.1928 |
| | | Autumn | 0.2970 | 0.2876 | 0.2982 | 0.2771 | 0.2599 | 0.1012 | 0.3064 | 0.2078 | 0.1850 | 0.1288 | 0.1321 |
| | | Winter | 0.2158 | 0.2054 | 0.2103 | 0.1900 | 0.1754 | 0.0663 | 0.2261 | 0.1524 | 0.1328 | 0.0877 | 0.0608 |
| | | Spring | 0.3383 | 0.3294 | 0.3429 | 0.3214 | 0.3028 | 0.1189 | 0.3473 | 0.2360 | 0.2115 | 0.1497 | 0.1470 |
| Ndeceleration (log cfu/g/day) [2] | | Summer | 0.1332 | 0.1311 | 0.1375 | 0.1310 | 0.1245 | 0.0496 | 0.1354 | 0.0924 | 0.0839 | 0.0613 | 0.0553 |
| | | Autumn | 0.1151 | 0.1115 | 0.1156 | 0.1074 | 0.1007 | 0.0392 | 0.1188 | 0.0805 | 0.0717 | 0.0499 | 0.0512 |
| | | Winter | 0.1008 | 0.0960 | 0.0983 | 0.0888 | 0.0820 | 0.0310 | 0.1057 | 0.0712 | 0.0620 | 0.0410 | 0.0284 |
| | | Spring | 0.1207 | 0.1175 | 0.1223 | 0.1146 | 0.1080 | 0.0424 | 0.1239 | 0.0842 | 0.0754 | 0.0534 | 0.0524 |
| Use-by date (days) [3] | | Summer | 12 | 12 | 12 | 12 | 14 | 34 | 12 | 19 | 20 | 27 | 30 |
| | | Autumn | 17 | 18 | 17 | 18 | 20 | 50 | 17 | 24 | 27 | 38 | 37 |
| | | Winter | 22 | 22 | 22 | 24 | 26 | 70 | 21 | 31 | 35 | 52 | 45 |
| | | Spring | 15 | 16 | 15 | 16 | 17 | 43 | 15 | 23 | 25 | 35 | 35 |

[1] *Ngrowth*— daily microbial population growth (log cfu/g) in the microbial growth (log) phase; [2] *Ndeceleration*— daily microbial population growth (log cfu/g) in the microbial deceleration phase; [3] the use-by dates for vacuum-packaged cooked sausages were established when spoilage microbial load achieved the maximum limit of ca. 9.3 log cfu/g. [4] PPCP—potentially postbiotic-containing preservative.

The temperature profile entered in the predictive model influenced the growth of the natural microbiota in vacuum-packaged cooked sausages. In summer, the values of the *Ngrowth* and *Ndeceleration* parameters of the predictive model, which represent the kinetics of the microbial growth in the growth (log) and deceleration phases, respectively, were higher than the values obtained during the other seasons. Thus, a shorter shelf life was observed during summer, with an early achievement of the predictive borderline limit, which can result in changes in sensory attributes related to sausages' appearance. As expected, microbial growth was reduced in winter. The correlation variable factor *FT*(*n*), which describes specific growth rates between log and deceleration phases, can also be used to indicate the impact of the temperature profile on microbial growth, highlighting that the critical period for sausage preservation was summer (*FT*(*n*) = 3.4894), followed by autumn (*FT*(*n*) = 2.8038), spring (*FT*(*n*) = 2.5801), and winter (*FT*(*n*) = 2.1401)

## 4. Discussion

Over the past decade, novel terms have been used to represent the beneficial effects of microorganisms. Postbiotics, or paraprobiotics or metabiotics, represent structural components of probiotic microorganisms and/or formulation of signaling molecules with a known chemical structure that can optimize host-specific physiological functions and regulate metabolic and/or behavior reactions related to the activity of host natural microbiota [28–30].

Hill et al. (2014) proposed that a more grammatically correct definition of probiotics would be 'live microorganisms that, when administered in adequate amounts, confer a health benefit on the host'. Thus, FAO and WHO definition of probiotics was reinforced as relevant and adaptable for current and further applications [31,32]. The development of metabolic by-products, dead microorganisms, or other microbial-based, nonviable products has potential; however, these do not fall under the probiotic construct'.

Once the viability of *Lacticaseibacillus* is reduced by cooking, postbiotic compounds can be a suitable alternative for the development of functional cooked foods. Moreover, precultured medium by LAB has been reported in the literature as a promising natural technology for food preservation [33].

Poor-quality raw material and inadequate handling can anticipate sausage spoilage. In vacuum-packaged cooked sausages, changes in the sensory attributes related to the sausages' appearance, which can be a decisive factor for consumer appraisal, occur when the microbial population achieves the stationary phase in the microbial growth curve (ca. 9.3 log cfu/g) (Figures S2 and S3, Table S1). The use of preservatives may reduce the activity of the natural microbiota, impacting the cell viability.

None of the treatments maintained the microbial load below the predicted model's borderline during the 90 days of shelf life indicated by the meat industry. Therefore, additional hurdles, such as cold storage, should be used combined with preservatives. When the cold storage temperature profile (7 °C) was entered in the predictive model to estimate the use-by date of the sausages, adding 3.0% of PPCP on the surface or adding 2.0% or more to the mass extended the use-by date by more than 90 days (Table 5).

**Table 5.** Durability study of vacuum-packaged cooked sausages stored at 7 °C.

| | Sample Incubation | | | Sausage Surface | | | | | | Sausage Mass | | | | |
|---|---|---|---|---|---|---|---|---|---|---|---|---|---|
| | Temperature (°C) | Time (days) | Blank | Control (water) 2.0% | Sodium lactate 2.0% | PPCP [4] 1.0% | PPCP 2.0% | PPCP 3.0% | Control (water) 2.0% | Sodium lactate 2.0% | PPCP 1.0% | PPCP 2.0% | PPCP 3.0% |
| Laboratorial data (log cfu/g) | | 0 | 5.77 | 5.80 | 5.71 | 5.88 | 5.79 | 5.80 | 5.66 | 5.76 | 5.81 | 5.83 | 5.87 |
| | 7 | 3 | 6.01 | 5.99 | 5.89 | 5.99 | 5.90 | 5.81 | 5.95 | 6.01 | 6.02 | 5.92 | 6.01 |
| | | 6 | 6.49 | 6.48 | 6.37 | 6.45 | 6.23 | 5.98 | 6.42 | 6.14 | 6.09 | 6 | 6.32 |
| | 36 | 2 | 6.69 | 6.72 | 6.70 | 6.80 | 6.64 | 6.14 | 6.59 | 6.66 | 6.61 | 6.33 | 6.26 |
| | | 4 | 8.50 | 8.48 | 8.49 | 8.61 | 8.45 | 6.87 | 8.42 | 7.12 | 7.10 | 6.97 | 6.94 |
| Specific maximum growth rate (log cfu/g/day) | 7 | L phase | 0.1000 | 0.0883 | 0.0850 | 0.0658 | 0.0550 | 0.0167 | 0.1117 | 0.0733 | 0.0583 | 0.0292 | 0.0608 |
| | | D phase | 0.0287 | 0.0253 | 0.0244 | 0.0189 | 0.0158 | 0.0048 | 0.0320 | 0.0210 | 0.0167 | 0.0084 | 0.0174 |
| | 36 | L phase | 0.5713 | 0.5650 | 0.5950 | 0.5713 | 0.5450 | 0.2188 | 0.5775 | 0.3950 | 0.3613 | 0.2675 | 0.2313 |
| | | D phase | 0.1637 | 0.1619 | 0.1705 | 0.1637 | 0.1562 | 0.0627 | 0.1655 | 0.1132 | 0.1035 | 0.0767 | 0.0663 |
| Ngrowth (log cfu/g/day) [1] | | | 0.1000 | 0.0883 | 0.0850 | 0.0658 | 0.0550 | 0.0167 | 0.1117 | 0.0733 | 0.0583 | 0.0292 | 0.0250 |
| Ndeceleration (log cfu/g/day) [2] | | | 0.0661 | 0.0584 | 0.0562 | 0.0435 | 0.0363 | 0.0110 | 0.0738 | 0.0485 | 0.0386 | 0.0193 | 0.0165 |
| Use-by date (days) [3] | | | 41 | 46 | 49 | 60 | 73 | 240 | 38 | 56 | 68 | 136 | 158 |

[1] *Ngrowth*—daily microbial population growth (log cfu/g) in the microbial growth (log) phase; [2] *Ndeceleration*—daily microbial population growth (log cfu/g) in the microbial deceleration phase; [3] the use-by dates for vacuum-packaged cooked sausages were established when spoilage microbial load achieved the maximum limit of ca. 9.3 log cfu/g. [4] PPCP—potentially postbiotic-containing preservative.

The addition of 2.0% of FCC85 on the surface or to the mass of the sausages little increased the use-by date. However, this is close to the maximum concentration permitted by the regulatory agency for the use of sodium lactate in heat-treated meat products [34]. This fact casts doubt on the efficacy of sodium lactate in increasing the use-by date of

vacuum-packaged cooked sausages. Although PPCP showed advantages compared with FCC85 regarding the extension of the use-by date of the sausages, it did not maintain the microbial load below the predictive model's borderline over 90 days either. However, there is no prescribed limit on the use of natural substances in sausages. Moreover, co-use of preservatives and proper management of the cold chain are suitable strategies to achieve a use-by date higher than 90 days.

Cold chain management of meat products, including raw material supply, processing, distribution, and retail, is a crucial factor to prevent spoilage [12]. The specific maximum growth rate obtained at 36 °C was expressively higher than the value determined at 7 °C (Table 2), showing the influence of the temperature on sausage spoilage. Indeed, the temperature profile during distribution, storage, and disposal in the market plays a role in the durability of meat products.

The addition of 3.0% of PPCP on the surface or 2.0% or more of PPCP to the mass, combined with management of the cold chain, resulted in a use-by date higher than 90 days (Table 5).

These results highlighted the potential use of PPCP on the surface of sausages. However, the concentration to achieve total inhibition of the microbiota, determined in vitro, should be respected. Thus, regarding the addition of PPCP to the mass of the sausages, the concentrations used to achieve partial inhibition of the microbiota can be used.

After packaging, syneresis may be induced during the storage and distribution of sausages, resulting in the accumulation of water, nutrients, and microorganisms inside the package. Preservatives are usually added to the mass with other ingredients during meat products preparation. However, there are no barriers to prevent microbial growth in the liquid accumulated inside the package after syneresis. Even when effective preservatives are added to the mass, this strategy may fail after syneresis because of the partial migration of these additives to the liquid phase. It can be of great concern if the storage temperature allows microbial activity.

The initial microbial load of the sausages may contribute to shortening the use-by date. By fixing the values of predictive model's parameters ($Ngrowth$, $Ndeceleration$, and $Ft(n)$) for each treatment, a use-by date of 90 days was achieved with the predicted initial microbial loads presented in Table 6.

**Table 6.** Estimated initial microbial load of vacuum-packaged cooked sausages to achieve the predictive model's borderline of 90 days.

| | | Presumed Initial Microbial Load (log cfu/g) | | | | |
|---|---|---|---|---|---|---|
| **Treatments** | | **Summer** | **Autumn** | **Winter** | **Spring** | **Cold Storage** |
| Blank | | −20.00 | −13.05 | −9.03 | −14.90 | 0.87 |
| Sausage surface | 2.0% of water (control) | −19.17 | −12.22 | −8.20 | −14.07 | 1.90 |
| | 2.0% of sodium lactate | −19.46 | −12.51 | −8.49 | −14.36 | 2.21 |
| | 1.0% of PPCP [1] | −17.59 | −10.64 | −6.62 | −12.49 | 3.88 |
| | 2.0% of PPCP | −16.38 | −9.43 | −5.41 | −11.28 | 4.84 |
| | 3.0% of PPCP | −6.57 | 0.38 | 4.40 | −1.47 | 8.30 |
| Sausage mass | 2.0% of water (control) | −20.91 | −13.96 | −9.94 | −15.81 | −0.24 |
| | 2.0% of sodium lactate | −14.21 | −7.26 | −3.24 | −9.11 | 3.26 |
| | 1.0% of PPCP | −12.46 | −5.51 | −1.49 | −7.36 | 4.56 |
| | 2.0% of PPCP | −7.62 | −0.67 | 3.35 | −2.52 | 7.18 |
| | 3.0% of PPCP | −6.27 | 0.68 | 4.70 | −1.17 | 7.57 |

[1] PPCP—potentially postbiotic-containing preservative.

Only the treatments with 3.0% of PPCP on the surface or 2.0% or more of PPCP in the mass of the sausages during the winter achieved the proposed use-by date. This result highlights the importance of considering additional factors to hurdle microbial growth in the sausages. During summer and spring, sausage preservation during the proposed use-by date was elusive for any treatment. With the co-use of preservatives and the management

of cold chain, the meat industry may reduce the initial microbial load to the levels presented in Table 6.

Satisfactory results regarding the extension of the shelf life of meat products can be achieved by reducing the initial microbial load, as well as by improving the product formulation to prevent syneresis [35]. Indeed, the microbial growth and durability of sausages are greatly influenced by the initial microbial load and the use of effective hurdles [36]. However, sporulated bacteria groups cannot be eliminated by cooking processes and hurdlers. This fact highlights the importance of avoiding the presence of these microorganisms in products by applying microbiological quality control in the meat supply chain [37].

Handlers, utensils, equipment, and microbial load of the raw material are the main microbial vehicles during production [38–40]. The environment is also a factor in meat spoilage [41], and it depends on the region; climate; microclimate, season; and anomalous environmental events such as forest fires, deforestation, rainwater excess, etc. [42].

## 5. Conclusions

PPCP produced by an axenic fermentation system with *L. paracasei* DTA 83 was as effective as the reference widespread commercial preservative FCC85 in preserving vacuum-packaged cooked sausages. Thus, it can be highlighted as a promising alternative concerning the use of natural technologies to preserve and produce functional cooked sausages. These results also revealed a logical relation regarding in vitro and in situ tests to evaluate sausage preservation. The concentration needed to achieve total inhibition of the microbiota, determined by an in vitro trial, should be respected when adding PPCP on sausages' surface. When adding PPCP to the mass of the sausages, the concentration that showed a partial inhibition in vitro could also be applied in situ. However, proper chain management during distribution and disposal of products in the market are pivotal to achieve the desired use-by date. Although this study presented a potential postbiotic alternative by adding PPCP to sausages, a robust in vivo trial must be further designed to evaluate effects in the host.

**Supplementary Materials:** The following supporting information can be downloaded at: https://www.mdpi.com/article/10.3390/fermentation8030106/s1, Figure S1: (a) Cluster analysis of RADP-PCR profiles obtained of 35 *Lacticaseibacillus* isolates from stool samples of infants aged between 7 to 21 days. The amplification patterns were analyzed using the software Gel Compar 4.1 (Applied Maths) [1], and (b) potential of *Lacticaseibacillus* to acidify the pasteurized deMan, Rogosa and Sharp broth medium.; Figure S2: The relative abundance (a and b), Krona plot (c), and dendrogram of similarities and discrepancies of high-throughput sequencing of bacterial phyla of vacuum-packaged cooked sausages; Figure S3: Microbial growth curves at 4 °C (a), 12 °C (b), 24 °C (**c**), and 36 °C (**d**). They were plotted regarding the natural microbiota of vacuum-packaged cooked sausages (⎯●⎯ sample #1\; ⎯◆⎯ sample #2\; ⎯△⎯ sample #3). Drop-plate technique was used to count total bacteria. Baranyi's mathematical model was applied to model the microbial growth at each temperature. The initial population was ca. 2.8 log cfu/g. The growth (log) phase started suddenly after incubation at 24 and 36 °C, and extended up to 8.2 log cfu/g. Stationary phase started after the population had reached ca. 9.3 log cfu/g. The period between the log and the stationary phases was considered the deceleration phase; Figure S4. Sample incubation design. Microbial count at time zero must be below 8.2 log cfu/g to validate the test. Besides the time zero, there is no predefined time for microbial counting once the computational predictive modeling can process any time; however, microbial growth (log) phase must be included at least in one of the counts. Laboratories can determine the incubation temperatures; however, lower and higher temperatures between 4 and 20 °C, 25 and 36 °C, respectively, must be used; Figure S5. Illustration of the biological growth curve by predictive modeling. A—adaptation and acceleration growth phase; L—microbial growth (log) phase; D—deceleration phase; S—stationary phase. Correlations between specific growth rate in L and D phases were performed based on the correlation factor FT(n) value, according to the chosen temperature profile of the test; Table S1. Instrumental color measurement (on the unopened packaged sausages and after withdrawing the sausages from the packages and cleaning up their surfaces) and slime formation detection [43,44].

**Author Contributions:** Conceptualization, A.L.d.L., C.A.G., L.M.C. and A.F.G.; validation, W.J.F.L.J.; writing—original draft preparation, A.F.G.; writing—review and editing, V.S.d.O.; supervision, R.H.L. and A.F.G. All authors have read and agreed to the published version of the manuscript.

**Funding:** This research received no external funding.

**Institutional Review Board Statement:** Not applicable.

**Informed Consent Statement:** Not applicable.

**Data Availability Statement:** Not applicable.

**Acknowledgments:** The authors are grateful to BRC Ingredients (Rio Claro, São Paulo) for its sponsorship.

**Conflicts of Interest:** The authors declare no conflict of interest.

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
