# Peer review of "A Natural Technology for Vacuum-Packaged Cooked Sausage Preservation with Potentially Postbiotic-Containing Preservative"

_fermentation, doi:10.3390/fermentation8030106_

Round 1
Reviewer 1 Report
The manuscript entitled “A natural technology for vacuum-packaged cooked sausage preservation with potentially postbiotic-containing preservative”, authored by de Lima, et al., will provide information for the natural preservation of cooked sausages. In the present study, the effect of potentially postbiotic-containing preservative produced by L. paracasei DTA 83 to extend the use-by date of vacuum-packaged cooked sausages was evaluated. The article is very interesting and the results seem conclusive. The manuscript still needs partial technical improvements.
- The author should be added the detail of biological replication.
- Line 72, first appearance the strain’s name in the main text, the full name required.
- Line 79, the genus name of the strain needs to be italicized.
- Line 247, delete the extra dot.
- Line 256, the “P” in “P < 0.05” should be italicized.
- Line 269, “increased in” should be “increased by”; Line 270, “22 days (winter)” should be “21 days (winter)”; The data in this sentence can’t be found in table 4 due to unclear presentation.
- “S0, S1, S2……” and “M0, M1, M2……” in Table 4, 5 and 6 are unclear.
- Some sentences are grammatically incorrect or inappropriate.
- Line 42, “among others” may be more appropriate to be replaced by “and others”;
- Line 48, “spoiling microorganisms” should be corrected to “spoilage microorganisms”;
- Line 221, “represent” can be replaced by “simulate”;
- Line 306-307, the sentence “This FAO/WHO requirement may difficult the development of cooked food matrices 306 containing probiotics, such as cooked sausages.” is grammatically wrong;
- Line 309-310 and line 391 “as a promisor” should be revised to “as a promising”;
- Line 317-318, the sentence “The meat industry prescribed the use-by date of 90 days for the vacuum packaged cooked sausages studied in this work” is grammatically wrong;
- Line 398, “is pivotal” should be “are pivotal”.
Author Response
Point 1: The author should be added the detail of biological replication.
Response: A paragraph was inserted (Lines 116 – 122) to provide the detail of biological replication.
Point 2: Line 72, first appearance the strain’s name in the main text, the full name required.
Response: It was modified accordingly.
Point 3: Line 79, the genus name of the strain needs to be italicized.
Response: It was modified accordingly.
Point 4: Line 247, delete the extra dot.
Response: It was modified accordingly.
Point 5: Line 256, the “P” in “P < 0.05” should be italicized.
Response: It was modified accordingly.
Point 6: Line 269, “increased in” should be “increased by”; Line 270, “22 days (winter)” should be “21 days (winter)”; The data in this sentence can’t be found in table 4 due to unclear presentation.
Response: It was modified accordingly (Line 269). In Line 270, the sentence was rephrased for a better understanding. Codes were replaced with a description of the treatment in Tables 4, 5, and 6.
Point 7: “S0, S1, S2……” and “M0, M1, M2……” in Table 4, 5 and 6 are unclear.
Response: Codes were replaced with a description of the treatment in Tables 4, 5, and 6.
Point 8: Some sentences are grammatically incorrect or inappropriate.
- Line 42, “among others” may be more appropriate to be replaced by “and others”;
- Line 48, “spoiling microorganisms” should be corrected to “spoilage microorganisms”;
- Line 221, “represent” can be replaced by “simulate”;
- Line 306-307, the sentence “This FAO/WHO requirement may difficult the development of cooked food matrices 306 containing probiotics, such as cooked sausages.” is grammatically wrong;
- Line 309-310 and line 391 “as a promisor” should be revised to “as a promising”;
- Line 317-318, the sentence “The meat industry prescribed the use-by date of 90 days for the vacuum packaged cooked sausages studied in this work” is grammatically wrong;
- Line 398, “is pivotal” should be “are pivotal”.
Response: Thanks for the contributions. The points were modified accordingly.

Reviewer 2 Report
- 47 - not every LAB is GRAS - this sentence can be misleading
- where does the maximum limit of ca. 9.3 log cfu / g, is it only based on prognostic studies, can some other scientific evidence support this?
- 211-219 - the temperature range has been established, but has the exposure time of the product been established until purchase? Can this be predicted in goals?
- 227- Results were expressed as Mean ± Standard Error (SE) from replicates - how many replications was this study carried out with?
- 138-139 - How was PPCP administered to the surface?
- I don't think I understood it, or I cannot find such information - against which microorganisms is the inhibitory effect determined by L. paracasei DTA 83? This seems to be important as not all microbes are negative in the production process. It is obvious that we will not get a sterile product that we can guide consumers, but some of them affect the taste and smell of meat products, including cooked ones.
- what is the hypothesis of this study? What the author wants to show through this experience should be clearly stated
- The greatest doubt is raised by the lack of a description of the rung used for bioconservation. Although the authors refer to other authors by citation, it is somewhat unsatisfied. What were the authors choosing this and not another strain? What is the expected effect (I know it is bacteriostatic due to metabolites, but have any studies been conducted in this area)? Has the strain been used before in meat products? if so - was it done without any quality losses? more information, e.g. in Introduction, should be given regarding L. paracasei DTA 83
Author Response
Point 1: 47 - not every LAB is GRAS - this sentence can be misleading
Response: Thanks for the contribution. The sentence was corrected.
Point 2: where does the maximum limit of ca. 9.3 log cfu / g, is it only based on prognostic studies, can some other scientific evidence support this?
Response: The literature reports that a ropy slime may be formed in meat products from 7 log cfu/g [1][2]. In the present study, the limit was based on a previous study that reported the occurrence of a ropy slime when the total microbiota achieved 9.3 log cfu/g, with the entrance of the microbiota in the stationary phase. Thus, the borderline was entered in the predictive model to predict when most consumers will refuse the sausages in the markets. The limit was determined in similar samples, produced in the same meat industry and with the same recipe and processing. The result will be presented in another paper. In the present study, Supplementary Material was provided to support such information. As numerous factors can influence microbial growth in sausages, the limit should be updated in each study.
- Borch, E.; Kant-Muermans, M.-L.; Blixt, Y. Bacterial spoilage of meat and cured meat products. Int. J. Food Microbiol. 1996, 33, 103–120, doi:https://doi.org/10.1016/0168-1605(96)01135-X.
- (BIOHAZ), E.P. on B.H. Growth of spoilage bacteria during storage and transport of meat. EFSA J. 2016, 14, e04523, doi:https://doi.org/10.2903/j.efsa.2016.4523.
Point 3: 211-219 - the temperature range has been established, but has the exposure time of the product been established until purchase? Can this be predicted in goals?
Response: The product was exposed for sale for 90 days or up to retaining its sensorial attribute related to appearance (consumer refusal due to ropy slime formation). The information was included in the study goals (Lines 98 - 100) and highlighted in the Discussion section (Line 397).
Point 4: 227- Results were expressed as Mean ± Standard Error (SE) from replicates - how many replications was this study carried out with?
Response: Results related to PPCP efficacy against the growth of natural microbiota and physico-chemical characterization of vacuum-packaged cooked sausages were obtained from triplicate. Each sample group in the in situ study was composed by n = 5. The Statistical Analyses section was rewritten to make it clearer (Lines 270 - 288).
Point 5: 138-139 - How was PPCP administered to the surface?
Response: It was based on the net weight of sausages in the packages. We have rewritten this point to make it clearer (Line 188).
Point 6: I don't think I understood it, or I cannot find such information - against which microorganisms is the inhibitory effect determined by L. paracasei DTA 83? This seems to be important as not all microbes are negative in the production process. It is obvious that we will not get a sterile product that we can guide consumers, but some of them affect the taste and smell of meat products, including cooked ones.
Response: Once the method to indicate changes in sensorial attributes related to the sausage appearance (ropy slime formation) was established based on total microbial growth, plate count agar was used for counting. This information was included in the text (Line 246).
Point 7: what is the hypothesis of this study? What the author wants to show through this experience should be clearly stated
Response: The objective of the study was rephased to clarify this point (Line 95 - 100).
Point 8: The greatest doubt is raised by the lack of a description of the rung used for bioconservation. Although the authors refer to other authors by citation, it is somewhat unsatisfied. What were the authors choosing this and not another strain? What is the expected effect (I know it is bacteriostatic due to metabolites, but have any studies been conducted in this area)? Has the strain been used before in meat products? if so - was it done without any quality losses? more information, e.g. in Introduction, should be given regarding L. paracasei DTA 83
Response: Figures S1(a) and (b) were included in the Supplementary Material to present preliminary results for choosing the DTA 83 in the revised version of the manuscript. It was based on the ability of DTA83 to assimilate sugar and produce acid in pasteurized broth medium compared to other lacticaseibacilli strains of DTA collection (IT/UFRRJ). We have inserted a paragraph in the Introduction section to show the bacteriostatic potential of lactic acid bacteria to inhibit the growth of the target microbiota and provide more information on DTA 83 (Lines 83 - 94).
PPCP was also produced by a semi-separately co-culture system with L. paracasei DTA 83 and Saccharomyces boulardii 17 to extend the shelf-life of vacuum-packaged cooked sausages. Losses of quality were not detected in the study. The paper was just approved in a special issue of Sustainability (MDPI journals).

Round 2
Reviewer 2 Report
Accept in present form